# Allometric equations for estimating above-ground biomass of *Nitraria sibirica* Pall. in Gobi Desert of Mongolia

**Javkhlan Nyamjav**[1], **Munkh-Erdene Batsaikhan**[1], **Guangliang Li**[2,3], **Jia Li**[4], **Amgalan Luvsanjamba**[5], **Kun Jin**[6], **Wenfa Xiao**[2,3], **Liji Wu**[7], **Tuvshintogtokh Indree**[1]*, **Aili Qin**[2,3]*

1 Laboratory of Vegetation Ecology and Plant Resources, Botanic Garden and Research Institute, Mongolian Academy of Sciences, Ulaanbaatar, Mongolia, 2 Research Institute of Forest Ecology, Environment and Protection, Chinese Academy of Forestry, Beijing, China, 3 Key Laboratory of Forest Ecology and Environment of National Forestry and Grassland Administration, Beijing, China, 4 Institute of Desertification Studies, Chinese Academy of Forestry, Beijing, China, 5 Gobi Bear Project, Ulaanbaatar, Mongolia, 6 Research Institute of Natural Protected Area, Chinese Academy of Forestry, Beijing, China, 7 Inner Mongolian Hulun Lake to National Nature Reserve, Hulunbuir, Beijing, China

* ailiqin521@163.com (AQ); i.tuvshintogtokh@gmail.com (TI)

**Data Availability Statement:** All relevant data are within the paper.

**Funding:** The study was funded by Mongolian Gobi Bear 356 Technical Assistance Project

## Abstract

*Nitraria sibirica* Pall. is a shrub species belonging to the family of Nitrariaceae. It plays pivotal role in arid ecosystems since it is tolerant to high salinity and drought. This species is widely distributed throughout Mongolia and it is mostly found in arid ecosystems of Mongolian Gobi Desert. In this study, we developed allometric equations for estimating above-ground biomass of *N. sibirica* using various structural descriptors and pinpointed the best models. Variables that precisely predicted above-ground biomass were a combination of basal diameter, crown area, and height. The allometric growth equation constructed is not merely helpful to achieve accurate estimations of the above-ground biomass in shrub vegetation in the Gobi Desert of Mongolia, but also can provide a reference for the above-ground biomass of *Nitraria* species growing in analogous habitats worldwide. Therefore, our research purposes an important advance for biomass estimation in Gobi ecosystems and complements previous studies of shrub biomass worldwide. This study provides reasonable estimates of biomass of *N. sibirica*, which will be valuable in evaluations of biological resources, especially for quantifying the main summer diet of Gobi bears, and also can be an alternative tool for assessing carbon cycling in Gobi Desert.

## Introduction

Plant biomass is an important indicator of ecological process and plays a key role in the ecosystem for its various purposes, such as estimations of net primary productivity, nutrient cycling, and wood production [1, 2]. Biomass estimation of grasses is relatively simple compared to woody plants, which have complex forms and its destructive harvest is time-consuming and costly [3]. Therefore, biomass estimation of shrubs is often neglected by researchers.

(2017400202306102) and the Mongolian Gobi Bear Technical Assistance Project of the Chinese Government.

**Competing interests:** The authors have declared that no competing interests exist.

Procedures for biomass estimation of shrub species consist of relating biomass components or total above-ground biomass to structural descriptors such as height (H), basal diameter (BD), crown area (CA), or volume (V) [4–8]. The estimation of total above-ground biomass is assured accurately when same independent variables are used for each component, the best fitting regression equations of each components are added, and regression coefficients of the individual biomass components are forced [9].

*Nitraria sibirica* Pall. is a shrub species belonging to the family of Nitrariaceae [10]. It contributes a major role in ecosystem due to its tolerance to high salinity and drought [11]. It is widely distributed in arid zones of the Near East from Central Asia to Northwestern China [12]. *N. sibirica* has been recorded in the most phytogeographical regions of Mongolia [13] and it is highly spread in arid ecosystems of Mongolian Gobi Desert [14]. According to the studies conducted on diet of Gobi bears (*Ursus arctos gobiensis*), Gobi bear is known to feed on fruits of *N. sibirica* in summer [15–18]. However, ineptitude of estimation on the above-ground biomass of this valuable species limits the integrity of ecosystem and habitat evaluations.

For the present study, we aimed to develop species-specific allometric equation to estimate the above-ground biomass of *N. sibirica*. The specific objective of this study is to develop the best-fitting allometric equations for *N. sibirica* through various biomass components, including branches, foliage, fruits and the total above-ground biomass using distinct structural descriptors such as height (H), crown area (CA), basal diameter (BD), and volume (V).

## Material and methods

### Study area

The study was carried out in the Great Gobi "A" Strictly Protected Area (GGSPA) in Trans-Altai Gobi of Mongolia which is located in the southwestern part of Mongolia (Fig 1). We have obtained permission to conduct field study in Great Gobi "A" Strictly Protected Area from Strictly Protected Area Authority of Ministry of Environment and Tourism, Mongolia. GGSPA was established as a protected area in 1975 and was designated as a UNESCO Biosphere Reserve in 1991 [16]. The highest peak is Tian Shan (2500–2700 m a.s.l) and the lowest altitude levels in the territory range between 700–1000 m a.s.l. [19].

The mean temperature is -7°C to -18°C (coldest day reaches -34°C) in winter and 25°C to 28°C (hottest day reaches 40°C) in summer [20]. According to our vegetation survey in 2017, 391 species belonging to 186 genera of 46 families have been recorded in GGSPA. Current field survey was conducted in all 3 oases (Tsagaan Bogd, Atas Inges and Shar Khulst) of GGSPA in spring and summer of 2019 and its geographical information is shown in Table 1. Each oasis consists of multiple water points where surface water exists, and artificial feeding boxes were established.

### Above-ground biomass sampling

A total of 8 plots (20m x 20m) were set up near water points and each plot was divided into 16 small quadrats (5m x 5m). For each selected individual, height (H), the crown diameter in two directions (largest and its perpendicular diameter of the crown), and basal diameter (BD) were measured. As a result, total of 35 individuals of *N. sibirica* were harvested using destructive method.

For biomass sampling, components including branches, foliage, fruits were harvested. The fresh weights of the samples were measured in the field using an electronic balance and all the samples were brought to the laboratory, oven-dried at 80°C to constant weight. Finally, the dry weight of each component was obtained. Total above-ground biomass was obtained by adding the biomass of branches, foliage and fruits. The main characteristics and biomass of *N. sibirica* are described in Table 2.

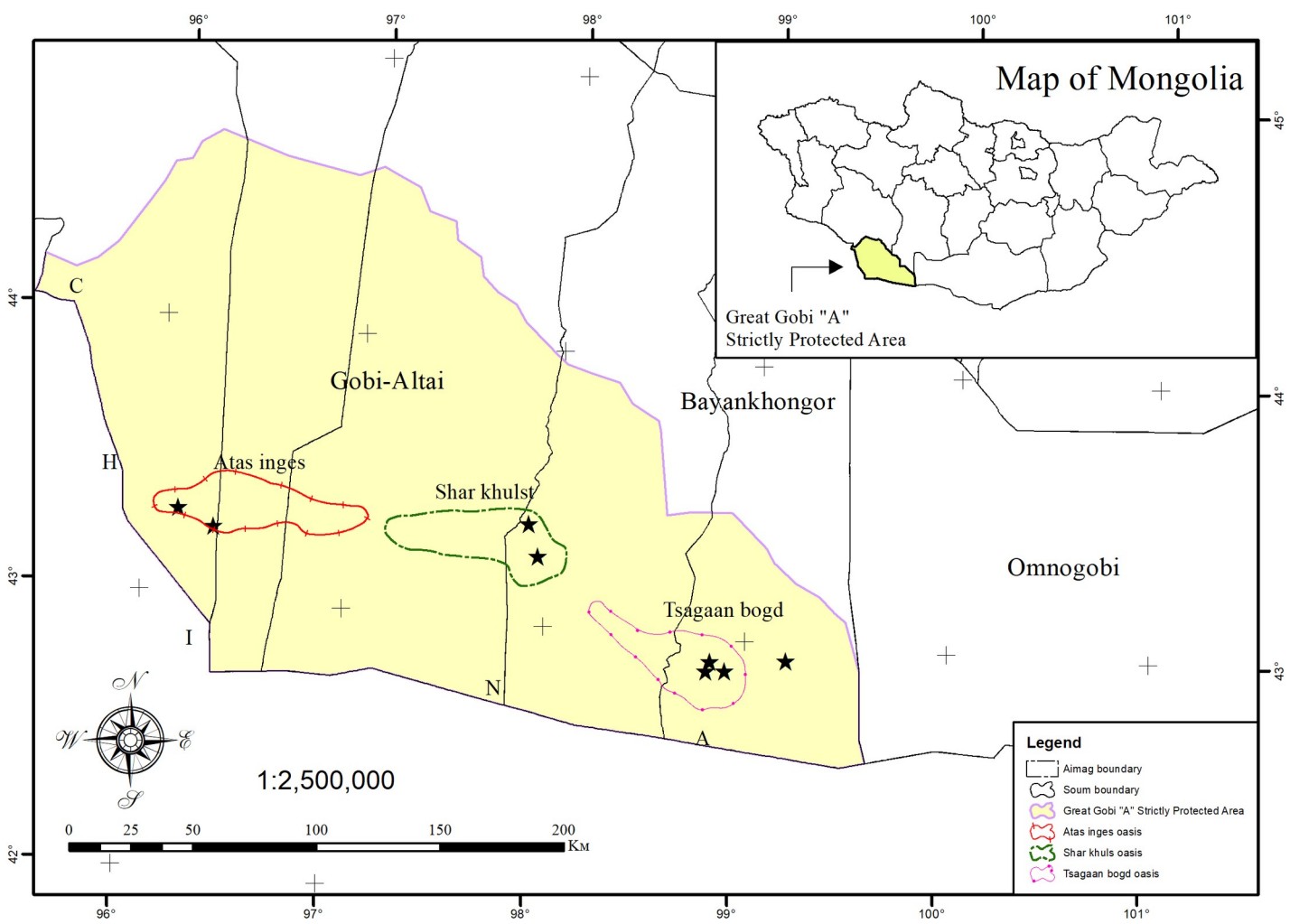

**Fig 1. Location of sampling plots in GGSPA of Trans-Altai Gobi, Mongolia.**

**Table 1. Geographical information of sampling sites in Great Gobi "A" Strictly Protected Area.**

| Site No. | Geographical location | Oasis name | Longitude (E) | Latitude (N) | Elevation (m) |
|---|---|---|---|---|---|
| 1 | Gobi-Altai aimag, Altai soum | Atas Inges | 96°9'1.05" | 43°18'36.752" | 1511 |
| 2 | Gobi-Altai aimag, Altai soum | Atas Inges | 96°20'7.135" | 43°15'6.177" | 1583 |
| 3 | Bayankhongor aimag, Bayan-Ondor soum | Shar Khulst | 97°53'2.685" | 43°21'33.27" | 1335 |
| 4 | Bayankhongor aimag, Bayan-Ondor soum | Shar Khulst | 97°55'59.181" | 43°14'59.319" | 1975 |
| 5 | Bayankhongor aimag, Shine jinst soum | Tsagaan Bogd | 98°50'19.498" | 42°55'54.6" | 1619 |
| 6 | Bayankhongor aimag, Shine jinst soum | Tsagaan Bogd | 98°49'36.376" | 42°52'17.863" | 1758 |
| 7 | Bayankhongor aimag, Shine jinst soum | Tsagaan Bogd | 98°54'40.105" | 42°52'58.013" | 1648 |
| 8 | Bayankhongor aimag, Shine jinst soum | Tsagaan Bogd | 99°12'11.683" | 42°56'3.827" | 1633 |

Note: Site No., Site Number.

**Table 2. Statistical characteristics and biomass of *N. sibirica* Pall.**

| Value | BD (cm) | H (cm) | Branch (kg) | Foliage (kg) | Fruit (kg) | Total AGB (kg) |
|---|---|---|---|---|---|---|
| Mean | 0.41 | 105.80 | 36.02 | 24.09 | 9.35 | 69.46 |
| SD | 0.14 | 42.74 | 38.50 | 29.63 | 24.02 | 84.11 |
| SE | 0.03 | 7.80 | 7.03 | 5.41 | 4.39 | 15.36 |
| Min | 0.19 | 40.00 | 0.33 | 0.21 | 0.10 | 0.56 |
| Max | 0.67 | 194.00 | 136.55 | 125.76 | 130.19 | 392.50 |

Note: BD, basal diameter; H, shrub height, AGB, above-ground biomass; Mean, arithmetic mean; SD, standard deviation; SE, standard error; Min, minimum value; Max, maximum value.

## Allometric equations

Single-variable and multiple-variable allometric equations were tested for estimating each component and the total AGB of *N. sibirica*.

$$\widehat{Y} = aCA^b \tag{1}$$

$$\widehat{Y} = aV^b \tag{2}$$

$$\widehat{Y} = aBD^b CA^c \tag{3}$$

$$\widehat{Y} = aBD^b V^c \tag{4}$$

$$\widehat{Y} = aCA^b H^c \tag{5}$$

$$\widehat{Y} = aBD^b CA^c H^d \tag{6}$$

Here, $\widehat{Y}$ is the predicted shrub biomass value in kg and single-variable refers to either crown area (CA), volume (V), or height (H) whereas multiple-variable refers to the combination of two or three of these variables. a, b, c and d are the fitted parameters. For single and multiple-variable equations, BD or H were not considered as primary variables, but as additional variables to improve the selected model.

Crown diameters were used to calculate crown area as follows:

$$CA = \frac{\pi \, x \, D_1 \, x \, D_2}{4} \tag{7}$$

where CA is crown area, $D_1$ is the largest diameter of the crown, $D_2$ is its perpendicular diameter. When estimating the biomass, log-transformed data is commonly used for linear regressions to eliminate the influences of heteroscedasticity [21, 22].

Therefore, Eqs (1) to (6) were linearized using logarithms in the following equations:

$$\widehat{lnY} = lna + b \, x \, lnCA \tag{8}$$

$$\widehat{lnY} = lna + b \, x \, lnV \tag{9}$$

$$\widehat{lnY} = lna + b \, x \, lnBD + c \, x \, lnCA \tag{10}$$

$$\ln\widehat{Y} = \ln a + b \text{ x } \ln BD + c \text{ x } \ln V \tag{11}$$

$$\ln\widehat{Y} = \ln a + b \text{ x } \ln CA + c \text{ x } \ln H \tag{12}$$

$$\ln\widehat{Y} = \ln a + b \text{ x } \ln BD + c \text{ x } \ln CA + d \text{ x } \ln H \tag{13}$$

where $\ln\widehat{Y}$ is the predicted shrub biomass value in the logarithmic unit and $\ln a$, b, c, and d, are the fitted parameters.

Log-transformed linear regression equations were frequently used for modelling above-ground shrub biomass in other studies [5, 6, 23]. Models were calculated separately for the branches, foliage, fruits and total AGB. A systematic bias could arise from the logarithmic transformation; thus, a correction factor (CF) was applied to correct the bias when back transforming the calculation [24]:

$$CF = \exp\left(RMSE^2/2\right) \tag{14}$$

where CF is the correction factor, and RMSE is the root mean square from the logarithmic regression. In order to select the best-fitting model for the total above-ground biomass and each biomass component, the coefficient of determination ($R^2$), RMSE, and Akaike Information Criterion (AICc) [25] were used.

$$R^2 = 1 - \left(\frac{\sum_{i=1}^{n}(\ln Y - \ln{-}Y)^2}{\sum_{i=1}^{n}(\ln Y - \ln\widehat{Y})^2}\right) \tag{15}$$

$$RMSE = \sqrt{\sum_{i=1}^{n}(\ln Y - \ln\widehat{Y})^2/(n - p - 1)} \tag{16}$$

$$AICc = n\log\left(\frac{RSS}{n}\right) + 2k + \frac{2k(k + 1)}{n - k - 1} \tag{17}$$

$$\Delta AICc_i = AICc_i - AICc_{min}, \text{ for } i = 1, 2 \ldots R \tag{18}$$

where $\ln Y$ is the observed log-transformed biomass value, $\ln\widehat{Y}$ is the predicted log-transformed biomass value from the fitted model, n is the sample size, $\ln{-}Y$ is the mean of the observed log-transformed biomass value, RSS is the residual sum of squares from the fitted model, k is the number of parameters, AICc is the Akaike information criteria, $\Delta AICc_i$ is the AICc difference, and $\Delta AICc_{min}$ is the minimum of the AICc values for the R models. The best-fitting model was selected according to the highest $R^2$ values and the lowest RMSE, AICc values. $\Delta AICc_{min} = 0$ indicates high model precision. Statistical analyses were carried out using the Minitabstatistical package.

## Results

Table 3 summarizes the equation parameters, accuracy, and goodness-of-fit for each of the 6 equations developed for *N. sibirica*. Single predictor variables such as BD, CD, H, CA, V were all tested to examine in which of them predicted biomass more specifically. Research findings showed that CA and V were the best single predictors. As a result, the highest $R^2$ value and lowest AICc suggest that basal diameter and crown area are the best predictors of above-ground biomass with the contribution of height. For branch, foliage, fruits, and total AGB,

**Table 3. Parameter estimates and model statistics of each model for branch, foliage, fruits, and total above-ground biomass of *Nitraria sibirica* Pall.**

| Component | Equation | lna | b | c | d | R² | RMSE | AICc | △AIC | CF |
|---|---|---|---|---|---|---|---|---|---|---|
| **Branch** | lnY = lna+b x ln(CA) | -12.39 | 1.404 | - | - | 0.768 | 0.905 | -1.050 | 0.279 | 1.506 |
| | lnY = lna+b x ln(V) | -13.96 | 1.097 | - | - | 0.675 | 1.07 | 1.862 | 3.191 | 1.773 |
| | lnY = lna+b x ln(BD) + c x ln (CA) | -13.01 | -0.468 | 1.469 | - | 0.775 | 0.917 | 0.275 | 0.004 | 1.523 |
| | lnY = lna+b x ln(BD) + c x ln (V) | -15.24 | 0.688 | 1.189 | - | 0.690 | 1.076 | 3.066 | 2.794 | 1.784 |
| | lnY = lna+b x ln(CA) + c x ln (H) | -11.95 | 1.425 | -0.151 | - | 0.769 | 0.929 | 0.504 | 0.232 | 1.540 |
| | **lnY = lna+b x ln(BD) + c x ln(CA) + d x ln (H)** | -12.85 | -0.451 | 1.473 | -0.045 | **0.775** | 0.945 | 2.071 | **0.000** | 1.563 |
| **Foliage** | lnY = lna+b x ln(CA) | -13.12 | 1.429 | - | - | 0.802 | 0.834 | -2.477 | 0.394 | 1.416 |
| | lnY = lna+b x ln(V) | -14.43 | 1.098 | - | - | 0.681 | 1.057 | 1.647 | 4.519 | 1.748 |
| | lnY = lna+b x ln(BD) + c x ln (CA) | -13.62 | -0.373 | 1.482 | - | 0.806 | 0.848 | -1.081 | 0.190 | 1.433 |
| | lnY = lna+b x ln(BD) + c x ln (V) | -15.44 | -0.542 | 1.171 | - | 0.690 | 1.072 | 2.997 | 4.269 | 1.776 |
| | lnY = lna+b x ln(CA) + c x ln (H) | -12.09 | 1.479 | -0.351 | - | 0.809 | 0.843 | -1.178 | 0.093 | 1.427 |
| | **lnY = lna+b x ln(BD) + c x ln(CA) + d x ln (H)** | -12.61 | -0.26 | 1.507 | -0.29 | **0.811** | 0.864 | 0.528 | **0.000** | 1.452 |
| **Fruit** | lnY = lna+b x ln(CA) | -23.19 | 2.134 | - | - | 0.540 | 2.309 | 15.220 | 1.454 | 14.379 |
| | lnY = lna+b x ln(V) | -27.71 | 1.807 | - | - | 0.558 | 2.265 | 14.885 | 1.119 | 13.002 |
| | lnY = lna+b x ln(BD) + c x ln (CA) | -25.3 | -1.59 | 2.358 | - | 0.566 | 2.309 | 16.324 | 0.958 | 14.379 |
| | lnY = lna+b x ln(BD) + c x ln (V) | -32.05 | -2.34 | 2.12 | - | 0.609 | 2.192 | 15.422 | 0.056 | 11.050 |
| | lnY = lna+b x ln(CA) + c x ln (H) | -26.74 | 1.964 | 1.21 | - | 0.564 | 2.313 | 16.354 | 0.988 | 14.512 |
| | **lnY = lna+b x ln(BD) + c x ln(CA) + d x ln (H)** | -31.31 | -2.27 | 2.209 | 1.74 | **0.611** | 2.252 | 17.166 | **0.000** | 12.626 |
| **Total AGB** | lnY = lna+b x ln(CA) | -12.44 | 1.464 | - | - | 0.794 | 0.876 | -1.620 | 0.242 | 1.468 |
| | lnY = lna+b x ln(V) | -14.06 | 1.143 | - | - | 0.697 | 1.062 | 1.725 | 3.588 | 1.758 |
| | lnY = lna+b x ln(BD) + c x ln (CA) | -12.99 | -0.415 | 1.523 | - | 0.799 | 0.889 | -0.250 | 0.013 | 1.485 |
| | lnY = lna+b x ln(BD) + c x ln (V) | -15.24 | -0.636 | 1.229 | - | 0.708 | 1.071 | 2.982 | 3.244 | 1.775 |
| | lnY = lna+b x ln(CA) + c x ln (H) | -11.94 | 1.488 | -0.169 | - | 0.795 | 0.898 | -0.083 | 0.180 | 1.497 |
| | **lnY = lna+b x ln(BD) + c x ln(CA) + d x ln (H)** | -12.71 | -0.384 | 1.529 | -0.079 | **0.799** | 0.916 | 1.537 | **0.000** | 1.521 |

Best equations with the highest R² and the lowest AICc values are highlighted in bold.

Eq 6 was selected as a good predictor of biomass. Consequently, the best-fitting equations for components and total AGB is shown in Table 4.

Biomass partitioning of *N. sibirica* was defined by branches taking the largest proportion of total biomass (52% on average), followed by foliage (35%), and fruits (13%) in Fig 2.

## Discussion

Above-ground biomass can be obtained by direct measurement [26], remote sensing techniques [27–29], and allometric equations [8, 30–32]. Direct measurement are costly and time consuming compared to remote sensing and allometric equation methods. Although remote sensing can provide multi-band and multi-temporal data sources for vegetation information extraction, it is difficult to detect the spectral information of the desert vegetation as desert

**Table 4. The best-fitting equations for components and total AGB of *Nitraria sibirica* Pall.**

| Components | Equations |
|---|---|
| **Branch** | AGB$_{est}$ = exp(-13.01)-0.468 x ln(BD)+1.469 x ln(CA) |
| **Foliage** | AGB$_{est}$ = exp(-12.61)-0.26 x ln(BD)+1.507 x ln(CA)-0.29 x ln(H) |
| **Fruits** | AGB$_{est}$ = exp(-31.31)-2.27 x ln(BD)+2.209 x ln(CA)+1.74 x ln(H) |
| **Total AGB** | AGB$_{est}$ = exp(-12.71)-0.384 x ln(BD)+1.529 x ln(CA)-0.079 x ln(H) |

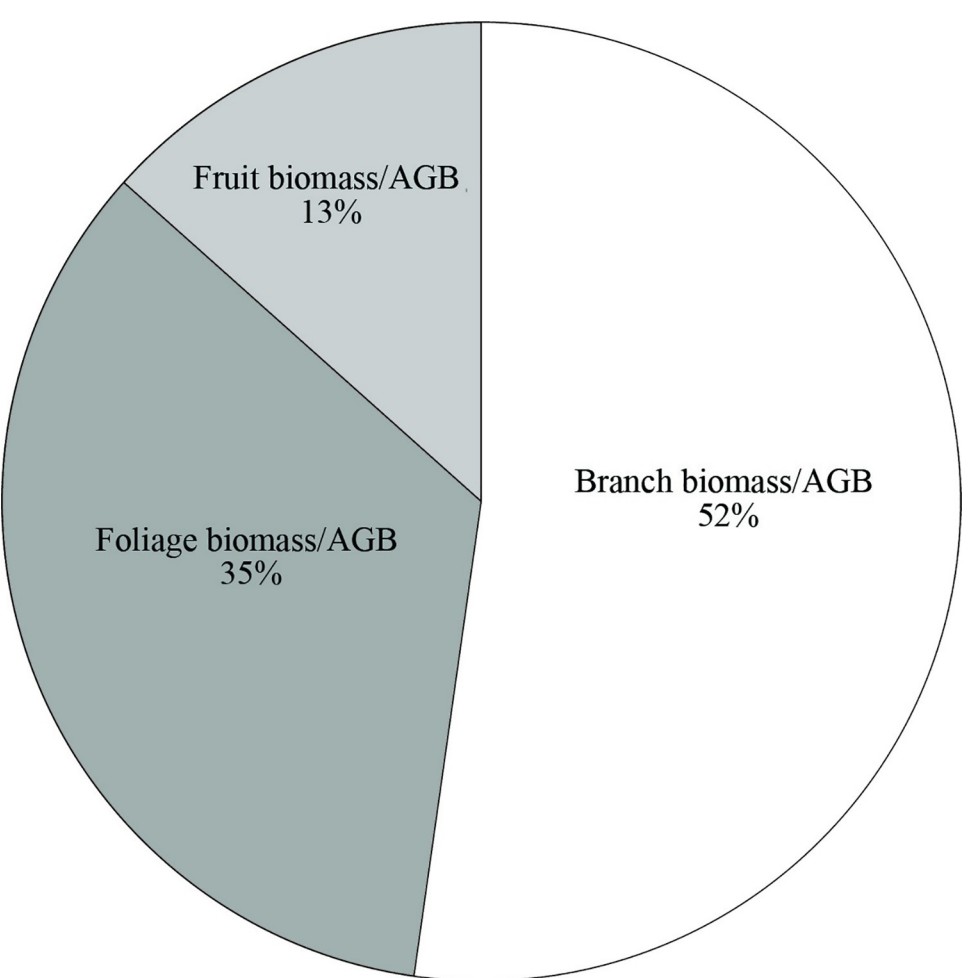

**Fig 2. Biomass partitioning of the above-ground components of *Nitraria sibirica* Pall.**

vegetation coverage is often less than 15% [33]. Therefore, allometric equation method is becoming the most pragmatic method to estimate [34] the above-ground biomass of species growing in the Gobi ecosystems. Simultaneously, allometric equations for estimating the above-ground biomass of shrubs continue to be limited in the literature compared to trees. In Mongolia, shrubs and biomass equations have been studied less, as a consequence, no literature has been published yet. Hence, we developed allometric equations for estimating above-ground biomass of *N. sibirica* using different variables and compared six models based on structural variables. Although all models offered good predictions of biomass, the best above-ground biomass model included basal diameter, crown area and height as independent variables (Eq 6).

Considerable amount of effort has been dedicated to develop allometric equations for estimation of shrub biomass worldwide, enclosing great range of species and regions including North America [4, 8, 35–38], South America [1, 7, 9], China [5, 6] and shrubs worldwide [23], on the other hand, semi-arid ecosystems have been disregarded, that led to limited studies of shrub biomass. For instance, Ali et al. [5] have found that diameter of the longest stem, height, wet basic density are best predictors, whereas Conti et al. [7] have chosen crown area and diameter of the longest stem as the best predictors of biomass estimation. In contrast to Ali

et al. [5], we found that D and H have comparatively poorer fit when used as a single independent variable. This difference is possibly due to the architectures of shrub species or environmental characteristics in the study ecosystems, as their study was done in subtropical forest, whereas this study was conducted in semi-arid ecosystem. Amount of literature studies showed that allometric equations can vary substantially from one region to another [39, 40].

According to the study conducted in semi-arid region similarly to our study by Conti et al. [23], basal diameter, crown diameter and height are the best indicators whereas volume and basal diameter were chosen as the best variables for estimating the AGB of shrub species by Yang et al. [6], and Zeng et al. [41], respectively. In agreement with previous studies [4, 6–9, 37, 42, 43] our results also selected basal diameter and crown area as reliable indicators for various biomass components of shrubs, furthermore we found that the addition of height improved the models. Consequently, it became evident that the future studies should consider the combination of BD with CA and H as independent predictor. Moreover, our results showed that H had a poor fit when used as a single predictive variable of AGB as consistent with [23] and although not many studies have tested the fit of V, our study recommends the use of V by virtue of its higher fit compatible with Yang et al. [6].

Finally, our findings indicate that AGB models incorporating a crown-related variable have significantly improved predictive power together with BD and H as the crown represents a relatively higher amount of biomass in shrubs compared to trees [23]. We also investigated biomass allocation of *N. sibirica* which resulted in branches making up the highest proportion followed by foliage and fruits.

In conclusion, our research purposes an important advance for biomass estimation in Gobi ecosystems and complements previous studies of shrub biomass worldwide. This study provides reasonable estimates of biomass of *N. sibirica*, which will be valuable in evaluations of biological resources, especially for quantifying the main summer diet of Gobi bears, and can also be an alternative tool for assessing carbon cycling in Gobi Desert. It is also an important manifestation of plant growth status and desertification detection. We will further provide tools for a methodological standardization of individual biomass quantification of *N. sibirica*. We presume that our results can contribute to the ecology by adding a novel biomass estimation model to previous biomass models across ecosystems.

## Acknowledgments

The authors would like to thank the team of the Great Gobi "A" Strictly Protected Area and Yun Wu (the Hulun Buir Municipal Committee Office) for helping through the field work. Also, we would like to express our gratitude to Dr. Altanzagas Batbaatar and Dr. Enkhmaa Erdenebileg, Botanic Garden and Research Institute, Mongolian Academy of Sciences and Dr. Brandon Bestelmeyer, USDA-ARS Range Management Research Unit, Jornada Experimental Range for recommendations which greatly improved the manuscript.

## Author Contributions

**Conceptualization:** Javkhlan Nyamjav, Amgalan Luvsanjamba, Kun Jin, Wenfa Xiao, Tuvshintogtokh Indree, Aili Qin.

**Formal analysis:** Javkhlan Nyamjav.

**Funding acquisition:** Kun Jin, Wenfa Xiao.

**Investigation:** Javkhlan Nyamjav, Munkh-Erdene Batsaikhan, Guangliang Li, Jia Li, Liji Wu, Aili Qin.

**Methodology:** Javkhlan Nyamjav, Tuvshintogtokh Indree, Aili Qin.

**Project administration:** Kun Jin, Wenfa Xiao.

**Supervision:** Kun Jin, Wenfa Xiao.

**Writing – original draft:** Javkhlan Nyamjav.

**Writing – review & editing:** Javkhlan Nyamjav, Aili Qin.

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
