## [Decision Letter · Decision Letter 0]

30 Jun 2020

PONE-D-20-13788

Allometric equations for estimating above-ground biomass of Nitraria sibirica Pall.
in Gobi Desert of Mongolia

PLOS ONE

Dear Dr. qin,

Thank you for submitting your manuscript to PLOS ONE. After careful consideration, we
feel that it has merit but does not fully meet PLOS ONE’s publication criteria as it
currently stands. Therefore, we invite you to submit a revised version of the
manuscript that addresses the points raised during the review process.

The paper lacks a proper discussion. You just re-dewscribe your results and compare
them with the results of other studies. However, they need to be put into a
reasonable biological context. Moreover, as you can see in the reviews, there are
also some technical issues that need to be mended.

I strongly recommend a linguistic check.

Please submit your revised manuscript by Aug 14 2020 11:59PM. If you will need more
time than this to complete your revisions, please reply to this message or contact
the journal office at plosone@plos.org. When
you're ready to submit your revision, log on to https://www.editorialmanager.com/pone/ and select the 'Submissions
Needing Revision' folder to locate your manuscript file.

If you would like to make changes to your financial disclosure, please include your
updated statement in your cover letter. Guidelines for resubmitting your figure
files are available below the reviewer comments at the end of this letter.

We look forward to receiving your revised manuscript.

Kind regards,

Dusan Gomory

Academic Editor

PLOS ONE

Journal Requirements:

2. In your Methods section, please provide additional location information, including
geographic coordinates for the data set if available.

4. We suggest you thoroughly copyedit your manuscript for language usage, spelling,
and grammar. If you do not know anyone who can help you do this, you may wish to
consider employing a professional scientific editing service.  

6. We note you have included a table to which you do not refer in the text of your
manuscript. Please ensure that you refer to Table 3 in your text; if accepted,
production will need this reference to link the reader to the Table.

Additional Editor Comments (if provided):

Reviewers' comments:

Reviewer's Responses to Questions

**Comments to the Author**

1. Is the manuscript technically sound, and do the data support the conclusions?

Reviewer #1: Yes

Reviewer #2: Yes

2. Has the statistical analysis been performed
appropriately and rigorously? 

Reviewer #1: Yes

Reviewer #2: Yes

3. Have the authors made all data underlying the
findings in their manuscript fully available?

Reviewer #1: No

Reviewer #2: Yes

4. Is the manuscript presented in an intelligible
fashion and written in standard English?

Reviewer #1: Yes

Reviewer #2: Yes

5. Review Comments to the Author

Reviewer #1: Dear Authors,

I read your paper carefully, and found any interesting study. I like the studies
related to allometric equations because it is much helpful for ecologist to estimate
biomass for diversity-functioning studies. The paper is generally well-written and
analyzed, but I believe that my comments will further help to improve the paper.

1) Strict English checking from a native speaker is important. MS lines were not
included and hence hard to mention the proper places for revision. Please replace
“ours were” by “our study was” or “this study was”. Fig. 3 replace “our equation” by
“this study”. Major revision needed in English.

2) Abstract: Research aim and conclusions are not very attractive. I suggest to write
based on knowledge gap, and what you have contributed to ecology.

3) Introduction: Very short paragraphs included. I suggest to merge small paragraphs
in to total of three paragraphs including the last one as it is.

4) M&M section is generally nice, but try to improve if you can.

5) Results: Table 2 can be merged in Table 1 by highlighting the best equations in
bold color with gray background. Write some details in the caption what represents
the best equation.

Fig. 3 is not very clear to me. Please use different relatively different colors for
symbols and also for regression lines. Hard to assess which equation is over or
under estimating the biomass. However, I think that Yang et al. is under estimating
the biomass whereas your equation is over estimating the biomass. So, Ali et al and
Conti et al are doing good as they the dots are near to the line but bit over
estimated as your equation. Have a look carefully.

6) Discussion. Major revision is needed here. What is the contribution of your study
to allometric biomass equations and ecology in general? Cite more relevant papers.
Go beyond the simple discussion. Say something broad. You have compared results with
three studies but those studies have recommended equations for species-specific and
multispecies equations. I think, you are comparing the multispecies equations with
your one species equation. So, there must be some differences. For example, in
discussion you mentioned that your results are different than Ali et al where they
found D, H and wood density as best predictors. So, you need to show further
explanation, why? Same for other two studies (yang et al. and conti et al.). I am
confused that why you are not comparing models results with Conti et al. 2015 (Ann
For Sci) which was conducted in semi-arid regions.

Good luck!

Arshad Ali

Reviewer #2: Biomass is an important component of global terrestrial ecosystem carbon
stocks. Here, the author developed a set of statistical models to estimate
above-ground biomass of N. sibirica. The results will improve our predictive ability
in aboveground biomass. However, the hypothesis, objective and rationale are not
clearly presented. For example, the author claimed that grasses destructive harvest
is difficult by citing Hughes et al., 1999. However, in Hughes’s paper, there is no
mention of the difficulties in grasses destructive harvest.

In addition, I have some suggestions for revision:

1. In the Material and methods section, all the field data are used to fit the
allometric model. Generally, in order to verify the power of the statistical model,
I suggest that you can use some data (eg. 75%) to estimate parameters and the rest
of data (eg. 25%) to verify the model.

2. In Figure 3, the x and y axis are all ranged from 0 to 25. But why is the 1:1 line
(intercept = 0) not a diagonal?

6. PLOS authors have the option to publish the peer
review history of their article (what does this mean?). If published, this will
include your full peer review and any attached files.

If you choose “no”, your identity will remain anonymous but your review may still be
made public.

**Do you want your identity to be public for this peer review?** For
information about this choice, including consent withdrawal, please see our
Privacy Policy.

Reviewer #1: **Yes: **Arshad Ali

Reviewer #2: **Yes: **Yan Li

---

## [Author Response · Author response to Decision Letter 0]

18 Aug 2020

I have responsed the reviewers in the file named"Response to reviewers"

to Reviewers.docx
---

## [Editor Report · Decision Letter 1]

3 Sep 2020

Allometric equations for estimating above-ground biomass of Nitraria sibirica Pall.
in Gobi Desert of Mongolia

PONE-D-20-13788R1

Dear Dr. qin,

We’re pleased to inform you that your manuscript has been judged scientifically
suitable for publication and will be formally accepted for publication once it meets
all outstanding technical requirements.

Kind regards,

Dusan Gomory

Academic Editor

PLOS ONE
---

## [Editor Report · Acceptance letter]

11 Sep 2020

PONE-D-20-13788R1 

Allometric equations for estimating above-ground biomass of *Nitraria
sibirica* Pall. in Gobi Desert of Mongolia 

Dear Dr. qin:

I'm pleased to inform you that your manuscript has been deemed suitable for
publication in PLOS ONE. Congratulations! Your manuscript is now with our production
department. 

Kind regards, 

on behalf of

Dr Dusan Gomory 

Academic Editor

PLOS ONE